# Streaming algorithms for evaluating noisy judges on unlabeled data - binary classification

## Abstract

The evaluation of noisy binary classifiers on unlabeled data is treated as a streaming task - given a data sketch of the decisions by an ensemble, estimate the true prevalence of the labels as well as each classifier's accuracy on them. Two fully algebraic evaluators are constructed to do this. Both are based on the assumption that the classifiers make independent errors on the test items. The first is based on majority voting. The second, the main contribution of the paper, is guaranteed to be correct for independent classifiers. But how do we know the classifiers are error independent on any given test? This principal/agent monitoring paradox is ameliorated by exploiting the failures of the independent evaluator to return sensible estimates. Some of these failures can be traced to producing algebraic versus real numbers while evaluating a finite test. A search for nearly error independent trios is empirically carried out on the `adult`, `mushroom`, and `two-norm` datasets by using these algebraic failure modes to reject potential evaluation ensembles as too correlated. At its final steps, the searches are refined by constructing a surface in evaluation space that must contain the true value point. The surface comes from considering the algebra of arbitrarily correlated classifiers and selecting a polynomial subset that is free of any correlation variables. Candidate evaluation ensembles are then rejected if their data sketches produce independent evaluation estimates that are too far from the constructed surface. The results produced by the surviving evaluation ensembles can sometimes be as good as 1%. But handling even small amounts of correlation remains a challenge. A Taylor expansion of the estimates produced when error independence is assumed but the classifiers are, in fact, slightly correlated helps clarify how the proposed independent evaluator has algebraic 'blind spots' of its own. They are points in evaluation space but the estimate of the independent evaluator has a sensitivity inversely proportional to the distance of the true point from them. How algebraic stream evaluation can and cannot help when done for safety or economic reasons is briefly discussed.

## 1 Introduction

Streaming algorithms compute sample statistics of a data stream. A *data sketch*, selected to fit the sample statistic one wants to compute, is updated every time a new item appears in the stream. A simple example of such a streaming algorithm is the use of two counters to compute the average value of a stream of numbers,

$$n, \mathrm{sum} = \sum_{i}^{n} x_i \tag{1}$$

$$\tag{2}$$

Submitted to 37th Conference on Neural Information Processing Systems (NeurIPS 2023). Do not distribute.

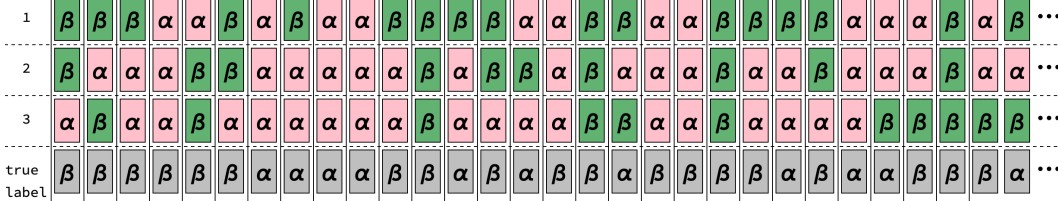

Figure 1: Stream of label predictions by three binary classifiers. Eight integer counters are enough to tally the number of times a particular prediction or voting pattern occurs when looking at *per-item* decision events.

The first integer counter tallies how many numbers have been observed so far. The second keeps a running total of the observed values. This data sketch is then used to compute the mean of the observed stream,

$$\mathrm{sum}/n. \tag{3}$$

Two things are notable about this simple algorithm. It only uses observed variables, and its computation is purely algebraic with them. There are no free parameters to tune or know beforehand. This paper discusses evaluation algorithms for binary classifiers that act similarly.

An evaluation on a finite set of labeled data is defined by the sample statistics it computes based on knowledge of the true labels. *Unlabeled evaluation*, the problem considered here, is the estimation of the values of these same sample statistics when the true labels are not known. This paper looks at how to do so when all we have are statistics of the decisions the members of an ensemble of noisy binary classifiers make - it is a 'black-box' algorithm. There are $2^n$ possible prediction events when we are concerned only about *per-item* evaluation statistics for $n$ binary classifiers. Figure 1 shows how an example of a stream labeled by three classifiers. To keep track of the decisions events at the per-item level, only 8 integer counters are needed.

For three binary classifiers, a *basic set of evaluation statistics* is defined by the prevalence of one of the labels, say label $\alpha$, $\hat{P}_\alpha$, and the label accuracies, $\hat{P}_{i,\alpha}$ and $\hat{P}_{i,\beta}$, for each of the classifiers. Note that these are *per item* statistics. They cannot quantify performance across items in the stream. Theorem 1 asserts that these variables are complete to explain the data sketches formed from the aligned decisions of independent classifiers. If we knew the value of the value of these evaluation statistics, we can predict exactly the value of the per-item data sketch counters. The challenge in unlabeled evaluation is to go the other way - to obtain estimates of the basic evaluation statistics starting from the data sketch.

Two evaluators for binary classifiers that are fully algebraic are built. The first is based on majority voting (MV). It decides what the correct answer key must be for the test. This makes it nearly impossible for it to return a correct answer even when its assumptions are satisfied. Even worse, it always provides seemingly correct estimates even when its assumptions are violated. The second evaluator is fully inferential. It never decides what the true label is for any items. Theorem 2 proves that this approach will correctly trap the true evaluation point to just two point candidates in evaluation space. Unlike the MV evaluator, it can return obviously incorrect estimates.

This paper assumes that these streaming evaluators are being deployed in an environment that has a principal/agent monitoring paradox. Evaluation ensembles, like decision ensembles, work best when they are independent in their error. Just as one would not want to incur the technical debt of having an ensemble of classifiers that always agreed, it makes little sense to deploy evaluation ensembles that are highly correlated. This raises two challenges when working on unlabeled data - how do we find these error independent evaluation trios, and how do we know, on any given evaluation, that they are still independent, or nearly so?

The approach taken here to ameliorate this monitoring paradox is that the failures of the independent evaluator can be used to exclude evaluation trios that are too correlated. The independent evaluator, by construction, is deterministic and always returns algebraic numbers. But its answers do not always make sense. We can detect this because we have prior knowledge about what *seemingly correct* evaluation estimates look like.

## 1.1 Properties of the true evaluation point

All the basic evaluation statistics are integer ratios by construction. For example, $\hat{P}_\alpha$, must be the ratio of two integers. Its numerator is some integer between 0 and the size of the test. The denominator, the size of the test. By construction, their ratio lies inside the unit interval. Similar considerations apply to any of the label accuracies for the classifiers - their true value must be an unknown integer ratio in the unit interval.

*Seemingly correct estimates* are estimated values that seem to be correct because they have this real, integer ratio form. Estimates that do not have this form are obviously incorrect. The naive evaluator built using majority voting by the ensemble always returns seemingly correct answers, never alerting its users that it is wrong and its evaluation assumptions do not apply on a given test.

The independent evaluator constructed from Theorems 1 and 2 is not like that. It fails, with varying degrees, to return seemingly correct estimates when the assumptions of a test are violated. The failures vary in their severity. The empirical hypothesis explored here is that their severity are indicative of the magnitude of the unknown decision correlations that are needed to correctly predict the observed data sketch.

## 1.2 A sample definition of decision error independence

Theorem 2 provides a closed, algebraic solution to the evaluation variety defined by the data sketch of error independent classifiers. The variety, the geometrical object in variable space that contains all points that satisfy a polynomial ideal, consists of just two points for these classifiers. But as noted above, independent evaluation ensembles are rare.

Handling correlation correctly requires that we introduce new evaluation variables to quantify it. Theorem 3 provides a a constructive proof of how to connect the data sketch of correlated classifiers to polynomials using these new correlation statistics plus those in the basic evaluation set. The evaluation variety, the set of values for the evaluation statistics that solves the polynomials, is not solved here for correlated classifiers. Nonetheless, it can be proven that the variety exists, its exact shape in evaluation space to be determined in future work. Nonetheless, a partial characterization of its shape is possible because of Theorem 3. The same process that solved the polynomial system for independent classifiers in Theorem 2 achieves a partial disentanglement of the variables when they are correlated. This defines a subset of the generating set that defines a surface computable without any knowledge of the correlations between the classifiers. This surface is not the evaluation variety but is guaranteed to contain it. This surface is used in the experiments to construct nearly independent evaluation ensembles.

## 1.3 Previous work and related topics

A mathematical treatment of the correctness of the decisions made when correct labels are assigned to majority voting dates back to Condorcet's analysis of the correctness of human juries during the French Revolution. But it was not until almost two centuries later that a mathematical treatment of using juries, this time human doctors, to evaluate themselves using only their aligned decisions was published by Dawid and Skene [1]. It proposed a probabilistic solution to evaluation by minimizing a likelihood function using the EM algorithm. This work was followed up in the early 2010s by a succession of papers in the NeurIPS conferences that took a Bayesian approach to constructing stream evaluator ([2], [3], [4], [5], [6]). Applications to evaluating workers in crowd-sourcing platforms was a big motivator for some of this research. The algebraic methodology proposed here would be economically impractical in such applications since it requires all the classifiers labeling every item in the stream.

The algebraic approach proposed here is closest to another probabilistic method, one proposed by Parisi et al. [7]. Rather than minimizing a likelihood function, it considers the spectral properties of matrices created by moments of the observed decisions. By hypothesizing hidden distributions, it then tries to carry out a matrix decomposition that eventually gives evaluation estimates. In contrast, the independent evaluator proposed here is purely algebraic. It invokes no assumptions about distributions. Nonetheless, Theorems 1 and 2 discussed here should be contrasted with Theorem 1 in the paper by Jaffe et al. [8], a solution for distribution-independent classifiers. The Supplement details the mathematical similarities and differences between the two solutions.

But research on direct evaluation seems to have waned since the 2010s. The 1st NeurIPS workshop on AI safety occurred last year and its RFP had a section on *monitoring* and *anomaly detection* that does not cite any of the above work [9]. Instead, research has focused on other aspects of monitoring that are important for AI safety - for example, carrying out risk minimization computations given unknown operating points for an ensemble as was done by Steinhardt et al. in [10].

Although stream algorithms do not usually calculate hidden knowledge statistics in a stream, some do. Good-Turing frequency smoothing [11] is one such algorithm. Its core intellectual idea - that you can estimate the probability of seeing hitherto unseen types of items using only the count for observed types - is used by LLMs whenever they want to estimate token sequences that were never observed during training.

Finally, most of the mathematical tools used in this paper come from *algebraic geometry* (AG) [12]. The use of algebraic concerns to study statistical problems was pioneered by Pistone et al. [13]. The topic is known as *algebraic statistics*. Like most of statistics, it is focused mostly on topics related to inferring distributions. This paper uses AG to estimate sample statistics.

## 2 Using majority voting to evaluate noisy classifiers

The correctness of decisions made by an ensemble that majority votes (MV) depends, roughly speaking, on two things. They must be error independent, and their labeling accuracies must be greater than 50% on each label. Item labels decided by majority voting will be correct more often than not if these conditions are met. This section details how a *naive* algebraic stream evaluator can be built on the basis of this decision algorithm. It works by imputing the correct labels or *answer key* for the observed items.

The integer counters of the per-item data sketch can be trivially turned into frequency variables. Each of the counters in a decision sketch tallies how often a decision event has been seen in the stream so far. There are only 8 decision events when considering the per-item decisions of three classifiers. The sum of their fractional frequencies, $f_{\ell_1,\ell_2,\ell_3}$, must sum to one,

$$f_{\alpha,\alpha,\alpha} + f_{\alpha,\alpha,\beta} + f_{\alpha,\beta,\alpha} + f_{\beta,\alpha,\alpha} + f_{\beta,\beta,\alpha} + f_{\beta,\alpha,\beta} + f_{\alpha,\beta,\beta} + f_{\beta,\beta,\beta} = 1. \tag{4}$$

The logic of MV evaluation is straightforward. The true label is given by MV, therefore the prevalence of a label is equal to the frequency that label was the majority vote. The estimate for the $\alpha$ label is thus a simple linear equation of these ensemble decision frequencies,

$$\hat{P}_{\alpha}^{(\text{MV})} = f_{\alpha,\alpha,\alpha} + f_{\alpha,\alpha,\beta} + f_{\alpha,\beta,\alpha} + f_{\beta,\alpha,\alpha}. \tag{5}$$

Similarly, we can write down algebraic formulas of the decision frequencies for each classifiers label accuracy. For classifier 1, those estimates are,

$$\hat{P}_{\alpha}^{(\text{MV})} = 1 - \frac{f_{\beta,\alpha,\alpha}}{f_{\alpha,\alpha,\alpha} + f_{\alpha,\alpha,\beta} + f_{\alpha,\beta,\alpha} + f_{\alpha,\beta,\alpha}} \tag{6}$$

$$\hat{P}_{\beta}^{(\text{MV})} = 1 - \frac{f_{\alpha,\beta,\beta}}{f_{\beta,\beta,\beta} + f_{\beta,\beta,\alpha} + f_{\beta,\alpha,\beta} + f_{\alpha,\beta,\beta}}. \tag{7}$$

The MV evaluator considers a classifier wrong if it votes against the majority.

These are algebraic functions of the frequencies derived from the data sketch, there are no free parameters. In addition, the estimates returned by them are always seemingly correct. All the MV estimates of prevalence and label accuracies are integer ratios inside the unit interval. The MV evaluator will never be able to alert its user that its assumptions are incorrect even when they are wildly off the mark.

### 2.1 The drawbacks of evaluating by deciding

Decision and inference are traditionally recognized as separate concerns in Machine Learning. Avoiding decisions and its hard choices until they are absolutely required typically leads to better performance. So it is here. Making a hard choice on the true label is going to be incorrect on some unknown fraction of the events that produced a particular voting pattern. Some of the times the

ensemble voted ($\alpha$, $\beta$, $\alpha$) it could have been a $\beta$item, not an $\alpha$one. This is expressed by the following equation,

$$n_{\ell_1,\ell_2,\ell_3} = \#(\ell_1, \ell_2, \ell_3 \mid \alpha) + \#(\ell_1, \ell_2, \ell_3 \mid \beta). \tag{8}$$

The number of times we saw a voting pattern is equal to the sum of times the items were $\alpha$ plus the times it was $\beta$. For any voting pattern by the ensemble, both labels are possible for any one item, no matter what the majority says. Zeroing out one term in this sum is an approximation. The supplement works out how this decision step means that the MV evaluator is hardly ever right even though it always seems so. Fixing this naive MV evaluator is easy - include both terms when expressing data sketch frequencies. Carrying out evaluation with these full equations is much harder but leads to an evaluator that is guaranteed to be correct when its assumptions are true.

# 3 Fully inferential evaluation of sample independent binary classifiers

Systems of equations can be wrong. Care must also be taken that they they define objects that exist so as to avoid making statements about non-existing entities. The two mathematical objects of concern here are systems of polynomial equations and the geometrical objects consisting of the points that solve them. The following theorem does this for error independent classifiers. It establishes that the basic evaluation statistics are sufficient to explain all observed data sketches created by them.

**Theorem 1.** *The* per-item *data sketch produced by independent classifiers is complete when expressed as polynomials of variables in the basic evaluation set.*

$$f_{\alpha,\alpha,\alpha} = \hat{P}_\alpha \hat{P}_{1,\alpha} \hat{P}_{2,\alpha} \hat{P}_{3,\alpha} + (1 - \hat{P}_\alpha)(1 - \hat{P}_{1,\beta})(1 - \hat{P}_{2,\beta})(1 - \hat{P}_{3,\beta}) \tag{9}$$

$$f_{\alpha,\alpha,\beta} = \hat{P}_\alpha \hat{P}_{1,\alpha} \hat{P}_{2,\alpha}(1 - \hat{P}_{3,\alpha}) + (1 - \hat{P}_\alpha)(1 - \hat{P}_{1,\beta})(1 - \hat{P}_{2,\beta})\hat{P}_{3,\beta} \tag{10}$$

$$f_{\alpha,\beta,\alpha} = \hat{P}_\alpha \hat{P}_{1,\alpha}(1 - \hat{P}_{2,\alpha})\hat{P}_{3,\alpha} + (1 - \hat{P}_\alpha)(1 - \hat{P}_{1,\beta})\hat{P}_{2,\beta}(1 - \hat{P}_{3,\beta}) \tag{11}$$

$$f_{\beta,\alpha,\alpha} = \hat{P}_\alpha(1 - \hat{P}_{1,\alpha})\hat{P}_{2,\alpha}\hat{P}_{3,\alpha} + (1 - \hat{P}_\alpha)\hat{P}_{1,\beta}(1 - \hat{P}_{2,\beta})(1 - \hat{P}_{3,\beta}) \tag{12}$$

$$f_{\beta,\beta,\alpha} = \hat{P}_\alpha(1 - \hat{P}_{1,\alpha})(1 - \hat{P}_{2,\alpha})\hat{P}_{3,\alpha} + (1 - \hat{P}_\alpha)\hat{P}_{1,\beta}\hat{P}_{2,\beta}(1 - \hat{P}_{3,\beta}) \tag{13}$$

$$f_{\beta,\alpha,\beta} = \hat{P}_\alpha(1 - \hat{P}_{1,\alpha})\hat{P}_{2,\alpha}(1 - \hat{P}_{3,\alpha}) + (1 - \hat{P}_\alpha)\hat{P}_{1,\beta}(1 - \hat{P}_{2,\beta})\hat{P}_{3,\beta} \tag{14}$$

$$f_{\alpha,\beta,\beta} = \hat{P}_\alpha \hat{P}_{1,\alpha}(1 - \hat{P}_{2,\alpha})(1 - \hat{P}_{3,\alpha}) + (1 - \hat{P}_\alpha)(1 - \hat{P}_{1,\beta})\hat{P}_{2,\beta}\hat{P}_{3,\beta} \tag{15}$$

$$f_{\beta,\beta,\beta} = \hat{P}_\alpha(1 - \hat{P}_{1,\alpha})(1 - \hat{P}_{2,\alpha})(1 - \hat{P}_{3,\alpha}) + (1 - \hat{P}_\alpha)\hat{P}_{1,\beta}\hat{P}_{2,\beta}\hat{P}_{3,\beta} \tag{16}$$

*These polynomial expressions of the data sketch form a generating set for a non-empty polynomial ideal, the* evaluation ideal. *The* evaluation variety*, the set of points that satisfy all the equations in the ideal is also non-empty and contains the true evaluation point.*

*Sketch of the proof.* The assumption that true labels exist for the stream items underlies the algebraic work required for the proof. The correct label of each item can be encoded in indicator functions, $\mathbb{1}_s(\ell)$, that are 1 if its argument is the correct label for item s, and zero otherwise. The existence of a true label for an item $s$ is then equivalent to the equation,

$$\mathbb{1}_s(\alpha) + \mathbb{1}_s(\beta) = 1. \tag{17}$$

Consider now the first term in Equation 8, $\#(\ell_1, \ell_2, \ell_3 \mid \alpha)$, as it relates to, say, the decisions event $(\alpha, \beta, \alpha)$. By using the predicted labels by the classifiers for a given items $s$, the following expression is exactly equal to one precisely at those decisions events but zero otherwise,

$$\mathbb{1}_s(\ell_{1,s})\,(1 - \mathbb{1}_s(\ell_{2,s}))\,\mathbb{1}_s(\ell_{3,s}). \tag{18}$$

The proof of having a complete representation using the basic evaluation variables then hinges in equating the average of this equation to the variables as follows,

$$\frac{1}{n_\alpha} \sum_{\mathbb{1}_s(\alpha)=1} \mathbb{1}_s(\ell_{1,s})\,(1 - \mathbb{1}_s(\ell_{2,s}))\,\mathbb{1}_s(\ell_{3,s}) = \hat{P}_\alpha\,\hat{P}_{1,\alpha}\,(1 - \hat{P}_{2,\alpha})\,\hat{P}_{3,\alpha}. \tag{19}$$

This equality is only true for independent classifiers because we have substituted the average of products of the indicator functions by products of their averages. New correlation variables are

Table 1: Algebraic evaluation formulas for the prevalence of $\alpha$, $\hat{P}_\alpha$, for three classifiers making independent errors on a test. The $\Delta_{i,j}$ and $f_{i,\beta}$ variables are polynomial functions of the data sketch frequency counters. Each $f_{i,\beta}$ is the frequency classifier 'i' voted for the $\beta$ label. The deltas are equal to $f_{i,j,\beta} - f_{i,\beta} f_{j,\beta}$, where $f_{i,j,\beta}$ is the frequency classifiers 'i' and 'j' voted simultaneously for the $\beta$ label.

| Evaluator | $\hat{P}_\alpha$ |
|---|---|
| Majority Voting | $f_{\alpha,\alpha,\alpha} + f_{\alpha,\alpha,\beta} + f_{\alpha,\beta,\alpha} + f_{\beta,\alpha,\alpha}$ |
| Fully inferential | $\frac{1}{2} - \frac{1}{2} \dfrac{(f_{\beta,\beta,\beta} - (f_{1,\beta} f_{2,\beta} f_{3,\beta} + f_{1,\beta} \Delta_{2,3} + f_{2,\beta} \Delta_{1,3} + f_{3,\beta} \Delta_{1,2}))}{\sqrt{4\,\Delta_{1,2}\,\Delta_{1,3}\,\Delta_{2,3} + (f_{\beta,\beta,\beta} - (f_{1,\beta} f_{2,\beta} f_{3,\beta} + f_{1,\beta} \Delta_{2,3} + f_{2,\beta} \Delta_{1,3} + f_{3,\beta} \Delta_{1,2}))^2}}$ |

introduced and then set to zero to define rigorously a sample definition of decision correlations. For example, the definition of the pair correlation variable on a label is given by,

$$\Gamma_{i,j,\ell} = \frac{1}{n_\ell} \sum_{\mathbb{1}_s(\ell)=1} (\mathbb{1}_s(\ell_{i,s}) - \hat{P}_{i,\ell})(\mathbb{1}_s(\ell_{j,s}) - \hat{P}_{j,\ell}) = \left( \frac{1}{n_\ell} \sum_{\mathbb{1}_s(\ell)=1} \mathbb{1}_s(\ell_{i,s})\, \mathbb{1}_s(\ell_{j,s}) \right) - \hat{P}_{i,\ell}\, \hat{P}_{j,\ell}. \tag{20}$$

Setting these pair correlations to zero then guarantees that we can write averages of the product of the indicator functions for two classifiers as the product of their label accuracies. Similar considerations apply to the product of the indicators for three classifiers. The consequence is that any decision event frequency by independent classifiers is complete when written in terms of the basic evaluation statistics. All data sketches produced by independent classifiers are predicted by the basic statistics. Since the proof is constructive and starts from expressions for the true evaluation point, we know that there is at least one point that satisfies all these polynomial equations. We conclude that the evaluation variety for independent classifiers exists and it contains the true evaluation point. □

Theorem 2 details exactly what the evaluation variety for independent classifiers must be.

**Theorem 2.** *The polynomial generating set for independent classifiers has an evaluation variety that has two points, one of which is the true evaluation point.*

*Sketch of the proof.* The quartic polynomials of the independent generating set are not trivial to handle. A strategy for solving them is to obtain algebraic consequences of them that isolate the variables. Using the tools of AG, this can be accomplished for independent classifiers. Solving their polynomial system is accomplished by calculating another representation of the evaluation ideal, called the Gröebner basis, that does this. It can be arranged to isolate the $\hat{P}_\alpha$ in a quadratic

$$a(\ldots)\hat{P}_\alpha^2 + b(\ldots)\hat{P}_\alpha + c(\ldots) = 0. \tag{21}$$

The coefficients a, b, and c are polynomials of the decision frequencies. Since this is a quadratic, it follows from the quadratic formula that it can only contain two solutions. This, coupled with the fact that other equations in the evaluation ideal are linear equations relating $\hat{P}_\alpha$ to $\hat{P}_{i,\alpha}$ or $\hat{P}_{i,\beta}$ variables leads one to conclude that only two points exist in the evaluation variety of independent classifiers. □

Table 1 compares the prevalence estimates of the fully inferential independent evaluator with the naive MV one. By construction, it will be exact when its assumptions apply. But unlike the naive MV evaluator, this formula can return obviously wrong estimates. The next section details how one can carry out experiments on unlabeled data to find evaluation ensembles that are going to be nearly independent.

## 4  Experiments with the failure modes of the independent stream evaluator

If perfect evaluations are not possible, one should prefer methods that alert us when they fail or their assumptions are incorrect. The experiments discussed here show how we can lever the self-alarming failures of the independent evaluator to reject highly correlated evaluation ensembles. There are four failure modes for the independent evaluator,

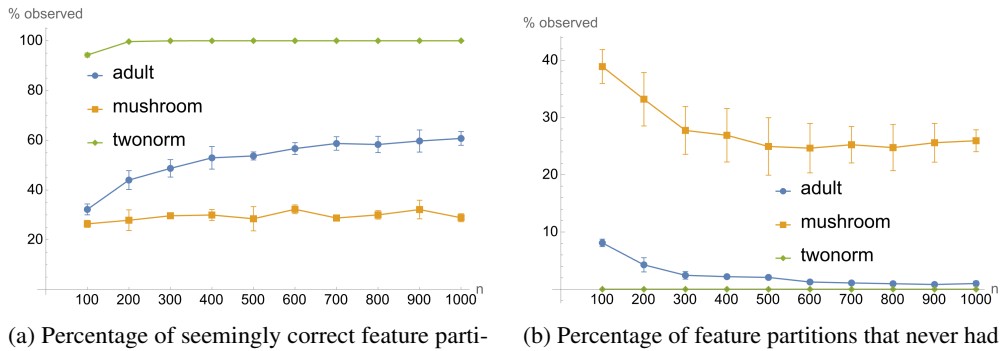

(a) Percentage of seemingly correct feature parti-
tions by test size.

(b) Percentage of feature partitions that never had
an independent model solution.

Figure 2: Failure rates for candidate evaluation ensembles constructed from disjoint partitions of the
features.

- The evaluation variety corresponding to the independent evaluation ideal is the empty set -
  no points in evaluation space can zero out the equations in the evaluation ideal.

- The two evaluation points contain complex numbers.

- The two evaluation points lie outside the real, unit cube.

- The estimated values contain unresolved square roots.

The fourth failure mode is interesting theoretically but not as practical. The Supplement details
how an unresolved square root in the evaluation estimates can be used to prove that the classifiers
were not error independent in the evaluation. Its theoretical interest lies in demonstrating that
algebraic numbers, unlike real ones, can be used to self-alarm, in an almost perfect fashion, when its
assumptions are violated.

The first three failure modes are more practical. The first set of experiments will look at how the
failures can be used to estimate what test sizes are less likely to have failed evaluations when we use
the independent evaluator. The second set of experiments profiles how well another rejection criteria,
this time based on the polynomial generating set for correlated classifiers, can help identify nearly
independent evaluations.

## 4.1 Rejecting highly correlated evaluation ensembles

Given a set of unlabeled data and a smaller set of labeled training data, the goal of the first set
of experiments is to construct and identify nearly independent evaluation ensembles on a larger,
unlabeled portion of data. This simple experimental set-up is meant to mimic a possible Auto-ML
application of stream evaluation. The experiments are meant to answer the question - how big should
an evaluation test be? This is done by profiling the algebraic failures as a function of test size.

Since the goal is to construct and then test if an evaluation ensemble is near enough independence to
give seemingly correct estimates, a generic training protocol was applied to the three datasets studied.
A training sample of 600 items was selected and the rest of the dataset was then held-out to carry out
unlabeled evaluations. The rate of failures on the held-out data is then used as a guide to select test
sizes that have low failure rates as observed empirically.

Independence in the candidate ensembles was maximized by training each member on features
disjoint with those used by the others. Disjoint partitions of the small 600 training set, each of size
200, were then used to train each candidate ensemble. A single profiling run selected 300 disjoint
feature partitions to test as the size of the held out data was changed. Averaging successive profiling
runs then gives an empirical measure of the failure rates as a function of test size. Each disjoint
feature partition was trained and evaluated ten times.

Figure 2 shows profiles of algebraic failures for the `adult`, `mushroom`, and `twonorm` dataset exper-
iments. Figure 2a plots the percentage of feature partitions that resulted in evaluation ensembles
returning seemingly correct estimates. The `twonorm` and `adult` experiments suggest that nearly
independent ensembles will be easier for them than in `mushroom`. Figure 2b plots the percentage of

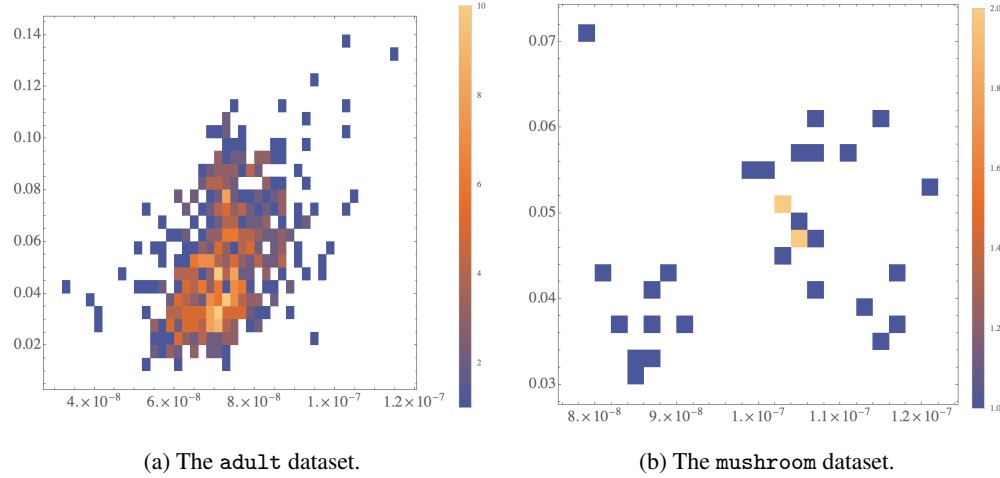

(a) The `adult` dataset.           (b) The `mushroom` dataset.

Figure 3: Maximum pair correlation in an evaluation ensemble versus the distance of the independent evaluator estimate it produces from the containing variety.

feature partitions that never produced data sketches explainable by the independent assumption. This plot also suggests that `mushroom` good evaluation ensembles will be harder to find.

## 5 A containing variety for arbitrarily correlated binary classifiers

The rejection of evaluation ensembles that have algebraic failures is no guarantee that the remaining ones will return somewhat accurate evaluation estimates. A different criteria has to be found to identify those that are close to independence. We can do this by considering the evaluation ideal of arbitrarily correlated classifiers.

Theorem 3 in the Supplement shows how a complete polynomial representation for their data sketch is possible when we include variables for each of the correlation statistics. Its corresponding evaluation variety remains an open problem. But another variety that contains it can be defined. And most importantly, it can be constructed without knowledge of the correlation statistics. Theorem 4 in the Supplement details how the basic evaluation statistics can be disentangled from the correlation ones by finding a suitable Gröbner basis for the generating set. The polynomials of the disentangled set have the forms,

$$\hat{P}_\alpha \left( \hat{P}_{i,\alpha} - f_{i,\alpha} \right) = (1 - \hat{P}_\alpha) \left( \hat{P}_{i,\beta} - f_{i,\beta} \right) \tag{22}$$

$$\left( \hat{P}_{i,\alpha} - f_{i,\alpha} \right) \left( \hat{P}_{j,\beta} - f_{j,\beta} \right) = \left( \hat{P}_{i,\beta} - f_{i,\beta} \right) \left( \hat{P}_{j,\alpha} - f_{j,\alpha} \right). \tag{23}$$

Since this generating set is a subset of the complete generating set for correlated classifiers, it is guaranteed to contain their evaluation variety, which, in turn, must contain the true evaluation point.

By self-consistency, if the evaluation ensemble was truly independent, it would be on this 4-dimensional surface. The second set of experiments looked at the hypothesis that data sketches from correlated classifiers would return independent evaluator estimates whose distance from the containing variety was related to their unknown amount of correlation. Figures 3a and 3b show the observed relation between the distance to the containing variety and an ensemble's maximum absolute pair correlation for the `adult` and `mushroom` experiments. As in the first experiments, the training data was 200 per label items. But the evaluation was carried out on held-out data with 2000 per label items. As was expected by the test size profiling runs, the `mushroom` seemingly correct evaluations were harder to find. The trend in these plots is suggestive but not conclusive.

The Supplement contains some of the evaluations from the least-distance evaluation ensembles. The evaluations on `twonorm` perform best, perhaps because that dataset is synthetic. But challenges remain when handling correlated ensembles. Perhaps these will be resolved with further work. This may be possible because we have a complete representation for correlated classifiers. Using that representation one can expand the independent evaluator estimates as Taylor series on the unknown

correlations. The linear term in $\Gamma_{i,j,\ell}$ has the inverse,

$$1/(\hat{P}_{k,\ell} - f_{k,\ell}). \tag{24}$$

Consequently, independent evaluator estimates becomes worse the closer one is to the "blindspots" in evaluation space at $f_{i,\ell}$. A look at the Gröbner basis for correlated classifiers shows how the blindspots shunt off the correlation variables by eliminating them from the basis. An evaluator whose evaluation statistics lie at the blindspots is thus unable to capture correlation effects - its sketch is explainable by an independent ensemble hypothesis that is not correct. This extreme case happens at a finite number of points, so its occurrence would be correspondingly rare. But as the Taylor expansion shows, it can affect the quality of the independent evaluator estimate severely if the true evaluation point lies near them.

## 6  Advantages and disadvantages of algebraic stream evaluation

The main advantage of algebraic evaluation is that it bypasses the representation and out-of-distribution problems in ML. Its focus is on estimating sample statistics with no concern for inferring models of the phenomena being classified or how the classifiers do it. There are no unknown unknowns in algebraic evaluation.

But algebraic evaluation cannot resolve the principal/agent monitoring paradox, only ameliorate it. Its batch approach only estimates average performance on a test. This may not be sufficient to handle anomalies or identify important subsets of the test where the classifiers perform much worse. In addition, sample statistics are not enough to identify the causes of poor performance or predict performance in the future. These are important considerations in settings one bothers to monitor with evaluation ensembles. Algebraic evaluators should be used in conjunction with other evaluation methods, such as the ones discussed in Section 1.3, that do encode more information about the application context.

Finally, all evaluation methods on unlabeled data are ambiguous. This is seen here by the two-point variety associated with independent classifiers. Additional assumptions about the evaluation must be made to 'decode' the true evaluation point. For example, in contexts where the prevalences are not expected to vary greatly their known value can be used to select to correct point. Such is the case in the `adult` dataset where the rare label corresponds to tax record features for people earning more than 50K US dollars annually. Fewer higher income records is a reasonable assumption for future random samples of US tax records. Conversely, if one could have high assurances of the quality of the classifiers and then use them to select the one point that aligns with it. In that case, stream evaluation is being used to monitor the environment and not the classifiers.

## 7  Broader Impacts

Evaluation on unlabeled data is a perennial problem in ML. As this conference and others discuss the impact AI agents have on our safety and society, it becomes necessary to have safeguards that can protect us from their decisions. The framework proposed here should have a positive impact across multiple application areas for ML since it is based on generic considerations.

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
