# OpenReview forum: "Streaming algorithms for evaluating noisy judges on unlabeled data - binary classification."
_NeurIPS.cc/2023/Conference — Submitted to NeurIPS 2023_

### Official Review · Reviewer_CtP8 · 2023-06-30

**Soundness:** 2 fair
**Presentation:** 2 fair
**Contribution:** 2 fair
**Rating:** 2
**Confidence:** 5

**Summary:**

The paper considers the problem of estimating the accuracy of noisy judges/classifiers in a streaming fashion, using only unlabeled data. Specifically, the goal is to compute the accuracy of each judge while processing items and the judge predictions for each item as part of a stream, without any associated labels for the items.

**Strengths:**

1. The problem of evaluating noisy judges that is being explored by the paper is an interesting and important one.

**Weaknesses:**

1. The main method presented in this paper (i.e., section 3) is not novel. For example, it can be seen as a special case of the approach presented in *Platanios, E. A., Blum A., and Mitchell T. "Estimating accuracy from unlabeled data" UAI (2014)*, which relaxes the independence assumption (though does not consider the streaming setting) and is also not cited in this paper. In fact, a lot of related work is missing including derivatives of the aforementioned paper (e.g., a direct follow-up in ICML 2016).

2. The premise of section 4 is weak. The idea that the method of section 3 is “self-alarming” because when you have dependent classifiers the accuracy estimates will be invalid is not completely correct. While this may capture some cases, there are still a lot of cases where you can have dependent classifiers, and where there exists a valid solution to the presented system of equations. Thus, I am not convinced by the main claim of this section.

3. The paper considers a streaming setting but it does not provide motivation for it. For example, it was not clear to me why we cannot store the predictions of the classifiers as items are processed in a database and then perform accuracy estimation periodically. If we have 8 classifiers and 2 possible labels, this would require 1MB per 1 million items, which does not seem expensive (and we can also perform random sampling if space becomes an issue).

4. The paper is presented in a manner that is hard to follow and could be significantly improved. I The whole paper would be presented in a simpler and more organized manner, but section 5 was particularly hard to follow without spending a significant amount of time to understand the argument that was being made.

5. The experimental evaluation is a bit lacking in that only toy datasets are being used, there is no explanation for what they are and why they are interesting, and there are very limited results being presented. Ideally, I’d like to see an “Experiments” section in the paper that describes the setup targeted at testing some hypotheses, the datasets, and the evaluation metrics, and then presents and discusses the evaluation results.

**Questions:**

1. Regarding the majority vote estimator, I don’t understand why you need 2^n variables in your sketch. I think all you need is n+1 variables which are defined as follows: (i) the number of items that have been processed thus far, and (ii) for each classifier, the number of times its predicted label matches the majority vote label. Is my understanding correct or am I missing something?

2. I didn’t understand section 2.1 and I also disagree with the statement in lines 163-164. I can see decision being framed as inference and I am also confident that oftentimes making hard decisions as opposed to keeping soft values around (which I assume is what you are referring to as “decision”) can be helpful. Can you please elaborate and also provide some reference for this claim if I am mistaken?

3. I didn’t understand what you mean by the “principal/agent monitoring paradox”. Can you please explain what that is and also why it is a paradox?

**Limitations:**

There is no discussion of limitations and potential negative social impact in this paper. One recommendation would be to try and think about what the implications could be for say voting systems, and also in situations where these methods are used to evaluate people whose income may depend on this evaluation (e.g., crowdworkers). In this case, the independence assumption being made by the paper may be too strong and yield in incorrect evaluations that could negatively and unfailrly affect the income of those people.

---

> ### Author Rebuttal · Authors · 2023-08-09
>
> We disagree strongly that Platonious et al. have already done this work and "better". As discussed in the general rebuttal, they are using the ensemble decisions PLUS additional logical constraints. Nowhere in their paper do they flag that three independent classifiers have a closed algebraic solution. Nor do they offer any theorems that their algorithm (which must be run for each evaluation) converges to the exact answer given its assumptions. In addition, they never discuss how their approach can detect the failure of its own assumptions.
> Question 1.  We did a quick calculation and interpreted your scheme as a linear operation from a 7 dimensional space (in the case of n=3) down to a four dimensional one (your four counters). How is a linear projection from 7 to 4 dimensions invertible? We disagree that n+1 counters can be used to keep a tally of 2^n independent events.
> Question 2.
> Question 3. The principal/agent monitoring paradox refers to the fact that the principal is forced to evaluate the work its agents to make sure everything is okay. But the principal delegated the work because they did not want to do it. In our work, the paradox arises because any evaluator that relies on unlabeled data and has no access to the ground truth can never be completely sure that it is carrying out a correct evaluation. The most one could do is ameliorate the paradox as we have done in this paper by discovering that algebraic evaluators have detectable failure modes. But no detector can be perfect, so this self-detection is not perfect. This is observed technically in our work in the well-defined blindspots that cause the evaluator to incorrectly return integer answers when, in fact, the classifiers are correlated. To state it succinctly - algebraic evaluation has no false positive rates on detecting correlation but has false negative cases when asserting that the classifiers are independent. This is not a terrible technical flaw because we can always detect when we are on the blindspot lines in evaluation space. So we can always flag the "on the blindspot" evaluations. For such evaluations, we cannot know if the classifiers are or are not independent.

---

### Official Review · Reviewer_7zjd · 2023-07-06

**Soundness:** 1 poor
**Presentation:** 1 poor
**Contribution:** 2 fair
**Rating:** 3
**Confidence:** 2

**Summary:**


This paper introduces a new inferential evaluator for evaluating noisy binary classifiers on unlabeled data in a streaming manner. Specifically, compared to the evaluator based on majority votes, the new evaluator gives a more complete and reasonable modeling of the true label prevalence and each classifier’s accuracy. In addition, the property of the new evaluator is also mathematically discussed, and the relationship between error dependence and the evaluator estimate is empirically discovered through experiments.

**Strengths:**


1. The paper addresses a significant problem in machine learning - evaluating the performance of binary classifiers on unlabeled data.
2. The proposed methods could have wide-ranging applications in various fields where machine learning is used, making the paper highly relevant.
3. The author provides mathematical proof to support the proposed methods and conducts empirical tests on several datasets.
4. The proposed generic framework that is based on algebraic geometry can cast a positive influence on evaluation methods on unlabeled data.
5.  The new algebraic evaluation method bypasses the representation and OOD problems in ML.

**Weaknesses:**

1. The paper is very hard to read and lacks the background to help the reviewer understand and improve the reading experience. Moreover, the supplement mentioned in lines 124, 171, 236, and 292 is missing from the paper. There are no detailed proofs for all the theorems.
2. The organization of the paper is confusing. The logic chain of the whole paper needs to be improved, better briefly introduce the outline and main content of each chapter at the beginning.
3. The aiming research problem needs to be explained formally in math language and to be explained clearly with intuitive explanations, better with a toy example or case study.
4. The current limitations of existing research, the proposed solutions (contributions of this paper), and the aiming experimental questions are not clearly listed, making it hard to catch the author's idea.
5. More experiments are needed. There are no experiments supporting that the performance of a majority vote-based evaluator is worse than that of the inferential one. And from the perspective of experiments, the advantage of the new evaluator is not made clear.
6. Only label prevalence is formalized in the paper, while there are no formulas for classifier accuracy.
7. The complexity of the concepts and the heavy use of mathematical proofs might affect their clarity. The author could consider providing more background information, intuitive explanations, or visual aids to improve the paper's accessibility.
8. The experiment part fails to show the superiority of the proposed method compared with other baselines.
9. The equations need to be carefully edited using formal math language. The space should be used for some meaningful and essential equations, not for simple ones such as summation or average operations.
10. Typos: In line 167, "it could have been a βitem, not an αone" should be "it could have been a β item, not an α one".

**Questions:**

    Is it possible to compare the proposed method to other baselines on the same datasets, in order to show the superiority and significance of the new method?

**Limitations:**

   The paper concludes with a brief discussion of how algebraic stream evaluation can and cannot help when done for safety or economic reasons.

---

> ### Author Rebuttal · Authors · 2023-08-09
>
> The experiments where meant to highlight what is unique about our algebraic evaluator - that it can detect when its own evaluation assumptions have failed. Since there is no other published evaluator on unlabeled data that can detect the failures of its own assumptions these experiments cannot have baseline comparisons. The very existence of the experiments and that an evaluator could ever detect the violation of its own assumptions is what makes them significant and, we believe, sufficient proof of the utility of this approach.

---

### Official Review · Reviewer_bq3o · 2023-07-15

**Soundness:** 3 good
**Presentation:** 2 fair
**Contribution:** 2 fair
**Rating:** 4
**Confidence:** 2

**Summary:**

This paper considers the problem of evaluating noisy binary classifiers on unlabeled streaming data. It aims to estimate the prevalence of the labels and the accuracy of each classifier on them, given a data sketch of label predictions by the members of an ensemble of noisy binary classifiers. The authors propose two algebraic evaluators based on the assumption of error-independent classifiers: the first is based on an additional assumption of majority voting, and the second is fully inferential and is guaranteed to contain the true evaluation point.

**Strengths:**

- The results of this paper seem to be well supported by the rigorous mathematical analysis as well as empirically demonstrated on three benchmark datasets.

**Weaknesses:**

- The independence assumption may not be satisfied in practice, especially when the ensemble of classifiers consists of different models trained on the same or overlapping datasets, or even the same model but trained for different durations.
- The significance of this paper is unclear. It would be better for the authors to provide some concrete real-world examples that fit the problem setting of this paper and explain the possible uses of the quantities desired to be estimated in these examples.
- The proposed evaluators may be sensitive to noise or corruption in the data sketch. The solutions of the algebraic equations will change if the data sketch is not perfectly recorded, leading to mistakes in distinguishing between independent and correlated evaluations.

**Questions:**

Please see Weaknesses.

**Limitations:**

The authors has discussed the limitations of this paper.

---

> ### Author Rebuttal · Authors · 2023-08-09
>
> 1. We agree with the reviewer that classifiers may not be independent. But as asserted in the paper, we consider the operational scenario where one engineers an evaluation ensemble that is nearly independent. What remains then, for safety reasons, is the ability to detect when the classifiers are not independent enough. This is why the paper focuses on the one novel aspect of our algebraic evaluator - that it can detect when the classifiers are not independent.
> 2. To make algebraic evaluation as proposed here fully operational, one would need to have the formulas that would allow you to measure the small correlations between nearly independent classifiers.
> 3. We are not sure why the mushroom and adult datasets are not examples of "real-world" data that illustrate concretely how to use algebraic evaluation. Since this is the foundational theoretical paper on this approach, we believed that focusing on just evaluation and not the implications of any particular evaluation was proper.
> 4. Yes, the proposed algorithm is not designed for adversarial situations where someone corrupts the input to the formulas. Most papers published in NeurIPS are also not immune to adversarial attacks. We do agree that the higher order polynomials involved in this work should be expected to have poor condition numbers in certain settings. Detailing these numerical sensitivities will be crucial in proving the reliability of future proposed algebraic formulas.

---

### Official Review · Reviewer_rucn · 2023-07-19

**Soundness:** 2 fair
**Presentation:** 1 poor
**Contribution:** 1 poor
**Rating:** 2
**Confidence:** 4

**Summary:**


The paper addresses making decisions based on the outputs of three binary classifiers. More precisely, it focuses on evaluating the performances of noisy classifiers. It considers majority voting on one hand, and a proposed evaluation scheme based on the classifiers' accuracies. The paper establishes several theorems so as to demonstrate the superiority of the second (proposed) evaluation scheme. Then, experiments are conducted to test the ability of the proposal to avoid making decisions in problematic situations—e.g., correlated classifiers.


**Strengths:**

I see no particular strength in the paper that would mitigate its flaws.

**Weaknesses:**

The proposal suffers from several major flaws.

First of all, it is badly written. The problem is not clearly stated. The mathematical objects (typically, the prevalence of the labels, and the label accuracies) are not properly introduced and defined. Some key notions (such as the "evaluation variety", the precise definition of correlated classifiers, among others) are also not defined. There are frequent references in the text to notions which have not been exposed yet (e.g., "the evaluators for binary classifiers" in the introduction, Theorems 1 and 2 in the introduction as well, Theorem 3 in Section 1.2).

Besides, the paper ignores a large amount of literature. The problem addressed has connections with computational social choice (voting schemes), of which some works are mentioned. But it also relates to classifier combination (boosting, error-correcting output codes, weighted averaging, racing algorithms, etc): the problem of evaluating the performance of an ensemble has been addressed in a number of works which are ignored here.

Last, but not least, the paper focuses on a very specific case—the ensemble has only three classifiers. This is very restrictive and such ensembles are hardly used in practice. Some claims are not supported—e.g., in Section 1.1, "Seemingly correct estimates are estimated values that seem to be correct because they have this real, integer ratio form. Estimates that do not have this form are obviously incorrect." This is not the case of the $F_\beta$ measure, for instance.


**Questions:**

How would your approach compare to classifier combination techniques where classifiers are combined based on their accuracy (e.g. racing algorithms or boosting) ?

How could the approach be generalized to more than 3 classifiers in the ensemble, or to multi-class classification problems ?



**Limitations:**

The authors have addressed the limitations of their approach, but in my opinion only in a restricted way. They have not addressed the potential negative societal impact of their work, but I do not think that this is crucial here.

---

> ### Author Rebuttal · Authors · 2023-08-09
>
> 1. We agree with the reviewer that this work has large applicability to the literature of social choice. We are also aware of the extensive work that has been done in other fields with this universal problem of principal/agent monitoring. That is precisely why we chose to frame the concerns of the paper within the principal/agent language. Given the size of the paper, we did not feel the difficulty of presenting this novel mathematical techniques gave us the luxury to discuss these wider concerns. We are also aware that there are many ensemble algorithms for decisions (such as boosting). How are these relevant to a paper that discusses using ensembles for evaluation, not decision?
> 2. The paper restricted itself to three classifiers because those are the only ones for which we have exact theorems. Research on ensembles of more than 3 classifiers remains open. We cannot tell, given our current knowledge, when the number of classifiers will become too large to solve. But note that if the purpose is just evaluation, and your classifiers are error independent, it does not matter how many you have. When one considers the technical debt of maintaining, and running multiple classifiers for the purpose of evaluation, three or four classifiers may be just right for a specific application.
> 3. Since this paper is concerned with evaluations, not combining the classifiers decisions to make a final, grand decision on each item we do not have any guidance on how to best combine their decisions. If one is concerned about the safety of one's decisions, it seems that the work on risk minimization by Hendrycks et al. would be the approach to take.

---

### Official Review · Reviewer_FuFc · 2023-07-27

**Soundness:** 2 fair
**Presentation:** 1 poor
**Contribution:** 2 fair
**Rating:** 2
**Confidence:** 2

**Summary:**

This paper considers the problem of evaluating an ensemble of binary classifiers on unlabeled data in a streaming setting. The authors first describe a baseline which treats the majority vote as the correct label and evaluate each classifier accordingly. Then they propose an evaluator based on an assumption that the classifiers are independent. The algebraic expression for this evaluator should return rational numbers if the assumption holds (as they should correspond to ratios of integer counts). Thus, this evaluator has failure modes that can be detected clearly unlike the majority-voting baseline that may return incorrect but seemingly sensible values.

**Strengths:**

While I am not an expert in evaluation using unlabeled data and cannot speak definitively, the proposed algebraic evaluation and characterizing failure modes by algebraic failures seem creative and novel.

**Weaknesses:**

I found the paper overall quite difficult to follow, and thus my assessment of its technical contributions may be limited. More thorough motivation and background on the problem setting (e.g. using real-world applications), as well as careful characterization of the proposed algebraic evaluator in contrast with existing approaches, would help make the paper much more approachable. I also had a hard time following most of Section 1 (especially 1.2) without the technical details in Section 3.

The paper is also missing some related work discussion, and its contributions with respect to prior work is not very clear. I struggled to see the connection to the works mentioned in the first paragraph of Section 1.3. Another work that appears very relevant is [1]; how does this paper relate to their approach?

Throughout the paper, only the setting with three binary classifiers is considered. A more general formulation may be helpful. As far as I can tell, this approach would scale exponentially in the number of classifiers which could limit its impact in practical settings.

Empirical evaluation was limited only to analyzing the failure rates, and there were no experiments on how well the proposed approach performs as an evaluator (i.e., how close are the error rate estimates to the true error rates?).

[1] Platanios, Emmanouil Antonios, Avrim Blum, and Tom Mitchell. "Estimating Accuracy from Unlabeled Data." 2014.

**Questions:**

Please see above for the main questions and suggestions. As a minor suggestion, I think Theorem 1 would be very intuitive to see using probabilities. It involves the probability of true label being $\alpha$ and the conditional probabilities of each outcome given the true label being $\alpha$ or $\beta$; the independence assumption allows turning the probability of a joint outcome into a product of probabilities for each classifier’s outcome.

**Limitations:**

Overall, yes. One limitation that was not discussed is that the approach seems to scale exponentially in the number of classifiers.

---

> ### Author Rebuttal · Authors · 2023-08-09
>
> 1. Relation of Platonious et al paper to this work. Their work is concerned with estimating the accuracy of multi-class classifiers whenever we have their decisions AND additional logical relations that can serve as ground truth constraints. Since whatever logical relations are chosen are specific to the particular evaluation (for example, the NELL-2 logical constraints in their Fig. 2), their work cannot be considered having universal application. By avoiding all possible constraints, our evaluator is truly universal in application. Furthermore, they never identify or assert that the case of three independent classifiers is solvable in a closed algebraic form. Their proposal is an algorithm that must be run for each evaluation. No such step is needed if one has a closed algebraic estimate.
> 2. The paper focused on three classifiers exclusively for succinctness, because we have exact theoretical proofs for such cases having solved the Groebner basis for three correlated and independent classifiers. We are currently working on a general solution to the Groebner basis for 4 correlated classifiers but have not succeeded. We know of no other work of evaluation on unlabeled data that can claim these exact solutions and use them as we have done here - to detect when the evaluation itself is failing.
> 3. We focused on measuring recognizable failure rates precisely because that is what is most novel about our approach. The appendix contains three evaluations for the datasets but it was not our main focus as you observe. We hope in future work to finish the research program started here and develop, perhaps with Taylor series expansions of the correlated Groebner basis, formulas that can give reasonable estimates when classifiers are weakly correlated.
> 4. Since the paper demonstrates that we can get exact solutions for the Groebner basis of independent and correlated classifiers in the case of three, we fail to why "exponential scaling" is a problem. Why do we need to use more than three or four classifiers? Are algorithms that do not scale well at large n useless at small n? Quicksort is not. We expect our exact solution will also be useful in many practical applications.

---

> > ### Comment · Reviewer_FuFc · 2023-08-20
> >
> > Thank you for your response.
> >
> > Regarding "why do we need to use more than three or four classifiers", I also didn't see any evidence in the paper or the author response to convince me otherwise. For general applicability, we would consider arbitrary numbers of classifiers, and as far as I can tell, the direct extension of the proposed method would scale exponentially in the number of classifiers. I do not see how quicksort algorithm is relevant here: it scales at worst quadratically, not exponentially.

---

> > > ### Author Response · Authors · 2023-08-21
> > > **Exponential nature of the problem versus our non-exponential solutions**
> > >
> > > Perhaps if we ground the discussion of exponential complexity in specific areas of the paper we can clarify our position that our proposed methods are not exponential and can be used with arbitrary number of classifiers. The problem of considering the voting patterns of an ensemble of binary classifiers is exponential by its very nature. There are 2^n possible voting patterns. And, as one of the theorems in the paper states, completely describing the data sketch of arbitrarily correlated classifiers requires that we know 2^(n+1)-1 sample statistics. But problems with exponential structures have non-exponential solutions. This paper shows two examples of this non-exponential solution to the problem of evaluating classifiers. The first one uses the independent solution for three classifiers to detect if they are so correlated that they return nonsensical solutions - to apply this method to ANY number of classifiers n>3 just requires that we compute the formula for all possible trios in an ensemble of n. This scales as n^3, a decidedly non-exponential solution to the problem of detecting highly correlated classifiers. In addition, we detail a universal (n+1) surface that applies to ANY ensembles of classifiers (correlated or not) in the (2n+1) evaluation space of the base statistics that only requires considering all possible pairs in an ensemble of n classifiers. This scales as n^2. Exactly as the Platanios agreement equations. Two methods to understand the evaluation of binary classifiers that are not, as the reviewer claims, scaling exponentially when applied to an arbitrary number of classifiers. It is not our fault that the nature of the problem of specifying the sample statistics of an evaluation by n binary classifiers requires 2^(n+1)-1. It is what it is. Any approach that completely describes the sample statistics of binary classifiers has to contend with that. This paper shows that the exponential nature of the problem is NOT a problem when it comes to completely solving independent classifiers. The exact solution shows that n=3 is SUFFICIENT. Absent some argument by the reviewer of why the exponential nature of the problem makes it NECESSARY to only consider exponentially hard solutions or that not doing so negates or invalidates the methods we have presented here, we really do not see the practical blocks the reviewer is arguing for. The correlation coefficients scale as 1/n. Since the only practical reason to have ensembles is to have them nearly independent, we see no reason why weakly correlated classifiers will not, in some future paper, be shown to be practically evaluated by just considering pair correlations or even triple ones. And the practical utility of this is no different than when one uses a finite number of terms in a Fourier series to approximate a function within a required error margin, or just use the first few terms in a Taylor series expansion.

---

### Author Rebuttal · Authors · 2023-08-09

This section will address two criticisms shared by two or more reviewers - 1. that the paper is not novel and the problem was treated already in Platonius et al. (2014) 2. the experimental results are weak and like baseline comparisons. The purpose of the paper was to devise an algebraic evaluator that has no free parameters, does not use probability and minimizes its evaluation assumptions. The utility of devising such an algorithm is that it would contribute to AI safety via the concept of defense in depth - providing another algorithm that can be used in conjunction with more complex and specific evaluators to provide multiple assessments of the performance of an AI system in production or in the field.
The catch-22 or paradox of evaluation on unlabeled data is that the evaluator itself is now suspect since it has to make assumptions. Any real evaluator, whether algebraic or probabilistic, will have them. How do we know these assumptions are correct during any particular evaluation on unlabeled data? Until this paper, no published paper has been able to show how this can be addressed. No paper in evaluation on unlabeled data known to us, not even Platonius et al. discuss how the evaluator can detect its own failure. No other paper in evaluation has shown that there is a closed, exact algebraic solution for three independent classifiers. This is not mentioned in Platonius et al or any other paper.
We do not view AI safety algorithm development as a horse race where one best algorithm should prevail. Rather, safety in depth requires that we consider multiple assesors to be truly safe. The algebraic approach here is not in competition with the approaches of Platonius et al or any of the ones we cite. Indeed, one of the technical achievements of the paper (not remarked on by any of the reviewers) is that we were able to solve the fully correlated 3-classifier Groebner basis. This allowed us to discover a n+1 dimensional surface inside the 2n+1 dimensional space of the unknown basic performance statistics. This is an immediate improvement that can be used by the methods of Platonious and others since it immediately restricts the possible correct answers.
Once one realizes that no other method has failure modes, the logic of the experiments presented should be clearer. A couple of reviewers criticized the paper becaused it lacked comparison baselines. How are we to tell how well algebraic evaluators are doing without these baselines? But no other method has a way of discovering the failure of its own assumptions so there can be no comparisons. Algebraic evaluators are unique in this feature and this is one of the major points of this paper and the subject of a theorem in the paper - we can detect, with no false positive rates, when the classifiers during a test did not act independently. But because no detector (even one of assumption failures) can be perfect, it is impossible to certify independence along well-designated blind spots that we detail in the paper.
AI safety cannot be predicated on algorithms that always succeed in the evaluations. This is impossible to attain. Tests will fail for reasons beyond the control of any evaluation algorithm. We are better off when we can detect such failures in a principled way that does not have fantastical claims that it will always work or never fail its own verification.
We share the reviewers wish for the completion of the full evaluation loop with this algebraic approach. Finding the unique ability to detects its own failure modes is useful but not sufficient. We need to develop the equations that will be able to detect small amounts of correlation when a system hovers around error independence. That remains an open reseach question we are working to address. In the appendix we showed three evaluations corresponding to the three datasets we used. Implicit in our work is the connection between failure modes and the size of the error correlations between the classifiers. Another open research question is if we can derive bounds on the correlation values when a failure occurs. For example, we have observed experimentally that correlations above 10% will trigger imaginary valued estimates. The attached PDF combines the experiments in the paper with the evaluations to show qualitatively how the evaluations improve as a dataset has less failure modes. This qualitative impression needs to be sharpened by developing numerical bounds that relate failure rates to correlation values.
Finally, a couple of reviewers viewed the paper's focus on just three classifiers as a weakness and not as a sign of the early stages of this research program. The strength of the paper lies in the exact theorems it is able to provide. These depend on the two major technical achievements of the paper - 1. the construction of an exact, algebraic solution for three independent classifiers and 2. the complete solution of the Groebner basis for three arbitrarily correlated classifiers. If the classifiers are independent, achievement 1 means that we have solved the evaluation problem for n >= 3. Just take any trio of classifiers and apply the exact algebraic solution. We are currently working on solving the correlated Groebner basis for four classifiers. This remains an open problem.
We may not have solved all the open research problems that this purely algebraic approach provides for those concerned by AI safety but we believe that its current achivements - detection of its own independence assumption and the (n+1) subsurface in the 2n+1 evaluation space are enough to complement other evaluation approaches.

---

> ### Author Response · Authors · 2023-08-19
> **Correction to Platanios reference**
>
> Our apologies for the mispelling of Platanios' name. In addition, the reviewers are correct that it was an omission not to reference their work. In particular, Platanios earlier work where the focus is on the algebra of agreement rates. We should have included another section in the Supplement detailing the mathematical relationship between their approach and ours. The summary is that their equations for agreement rates are completely linear, thus avoiding the mathematical complexities of the quartic polynomials we considered. As one would expect, these the linear agreement equations of Platanios, carry less information than the full polynomials. One possible similarity between Platanios' agreement equations and our work could be the generating polynomials set for the (n+1) surface we identify that is universal since it involves a subset of polynomials that involve pairs of classifiers. But our equations are quadratic, still not linear as in Platanios' agreement equations. It is inevitable that any probabilistic approach, which is the route continued by Platanios, should contain algebraic constraints and equations similar to ours. What distinguishes our work is it strict adherence to only algebraic assumptions so as to pare down the assumptions necessary for empirical evaluation to a bare minimum, thus providing a different corroboration that an AI system is operating safely.

---

> ### Author Response · Authors · 2023-08-22
> **Platanios agreement equations are incorrect**
>
> As we were elaborating the similarities between our approach and Platanios' agreement equations, we discovered that the independent solution presented in his work with Tom Mitchell is mathematically incorrect. The agreement rate of independent classifiers ACROSS labels is not equal to the product of their individual error rates ACROSS labels. This is only true, INTRA label. We plan to update our supplement with an algebraic demonstration of their logical mistake as well as a numerical demonstration that the incorrect solution fails to predict the correct error rate for individual classifiers.

---

### Decision · Program_Chairs · 2023-09-21

**Decision:**

Reject

**Comment:**

There are promising ideas in this paper, and most of the reviewers agreed that the problem being studied is important and interesting. We also appreciate the authors’ efforts in providing detailed responses. However, it seems that the current presentation of the paper has limited the ability of readers to see the value in the work and to put it in context of prior work. Hopefully, the feedback and process of writing responses will help in improving the paper for a future submission that is more valuable to a general ML audience.